# OpenReview forum: "Comparing the Evaluation and Production of Loophole Behavior in Humans and Large Language Models"
_EMNLP/2023/Conference — EMNLP 2023 Findings_

### Official Review · Reviewer_HszC · 2023-07-28

**Soundness:** 2

**Excitement:**

3: Ambivalent: It has merits (e.g., it reports state-of-the-art results, the idea is nice), but there are key weaknesses (e.g., it describes incremental work), and it can significantly benefit from another round of revision. However, I won't object to accepting it if my co-reviewers champion it.

**Paper Topic And Main Contributions:**

This paper focuses on the evaluation and production of loophole behavior. Here, the loophole is the situation when people understand the intended meaning or goal of another person, but choose to go with a different interpretation. In this paper, the authors analyze to what extent LLMs capture the pragmatic understanding required for engaging in loopholes.

The main contribution of this paper is the NLP experiment, when the performance of five SOTA LLMs is examined and compared to humans on two tasks: evaluation (ratings of trouble, upset, and humor), and generation (coming up with new loopholes in a given context). In the evaluation task, the findings are that while many of the models rate loophole behaviors as resulting in less trouble and upset than outright non-compliance (in line with humans), they struggle to recognize the humor in the creative exploitation of loopholes in the way that humans do. In the generation task, only two of the models were found capable of reliably generating loopholes of their own.

**Reasons To Accept:**

This paper is a good addition to the dispute on whether and how well LLMs do at pragmatic language use. It helps better understand the scope and limits of pragmatic reasoning abilities in LLMs and helps address AI safety concerns. As such, the work is of interest for the NLP community.

**Reasons To Reject:**

I can see several concerns related to the soundness of methodology, and as a result, the strength of claims in this paper.
1. More details need to be added to the description of how human evaluation was performed - was each scenario evaluated by a single or multiple humans? What was the level of inter-rater agreement? Are there any differences in evaluation results across genders, ages or backgrounds? This information could provide additional insights on the results of "funny" and "upset" ratings. It is well known that both humor and distress are strongly influenced by culture (see Martin, R. A., and Ford, T. (2018). The Psychology of Humor: An Integrative Approach. Burlington, MA: Elsevier Academic Press, also Hashimoto, T., Mojaverian, T., & Kim, H. S. (2012). Culture, interpersonal stress, and psychological distress. Journal of Cross-Cultural Psychology, 43(4), 527-532), and this could be related to the finding that fewer models are able to differentiate behavior types when it comes to reasoning about humor and distress.
2. There is no explanation of what is the underlying motivation for selecting these specific models for this analysis. It would be interetsting and useful to understand how the evaluation and generation results differ depending on the model characteristics, such as e.g. size and specifics of the training/fine-tuning process. Without such analysis, it is not clear whether any aspects of loophole behavior have a potential to generalize to the other LLMs, and this reduces the value of the work.
3. There is no reported statistical analysis of the presented results in the paper and this makes it difficult to properly interepret most of the findings. Many claims sound unsupported due to this, such as e.g. "only three of the models, GPT-3.5, GPT-3, and Tk-Instruct-3B reflect human baselines on the metric of how upset the other person would be" (lines 605-607). Visually, it looks like at least FlanT5-XL and FlanT5-XXL also "reflect" human ratings if by "reflect" the authors mean the same sorting order. The claim "the advantages of increasing model size to improve pragmatic reasoning abilities are apparent" (lines 617-619) is not supported by the findings either - for the ratings of "upset" Tk-Instruct-11B is performing worse than Tk-Instruct-3B, for the "trouble" rating InstructGPT (175B) is performing worse than FlanT5-XL (3B), for the generation task TkInstruct-3B is performing better than FlanT5-XXL of the size 11B, etc.

**Reproducibility:**

3: Could reproduce the results with some difficulty. The settings of parameters are underspecified or subjectively determined; the training/evaluation data are not widely available.

**Reviewer Confidence:**

4: Quite sure. I tried to check the important points carefully. It's unlikely, though conceivable, that I missed something that should affect my ratings.

---

> ### Author Rebuttal · Authors · 2023-08-28
>
> Thank you for taking the time to engage with our work. We were glad to hear you find this work of potential interest to the NLP community and that you find it a good addition to the current discussion on pragmatic language use in LLMs. We also greatly appreciate your feedback, and the opportunity to address your concerns.
>
> --
>
> Point 1, further details on human evaluation: For the evaluation task human baseline, 180 participants evaluated all of our 36 scenarios (so 180 participants per scenario). For each scenario, participants were randomly assigned to compliance, loophole, or non-compliance behavior. Therefore, for each participant, 12 scenarios would describe loophole behavior, 12 compliance, and 12 non-compliance, counterbalanced across participants. This procedure for collecting the human baseline results in robust population-level effects, and calculating inter-rater agreement in such a setting is not typical.
>
> An exploratory analysis breaking down our present results by gender (Male, Female, Non-binary) and age groups (18-30, 31-40, 41-50, 51-60, 61-70, 71-80), reveals consistent patterns of funny, upset, and trouble. These results represent robust population-level effects, with 180 participants evaluating all 36 scenarios, and have additionally been replicated in multiple studies (Parece et al. “Skirting the Sacred: Moral Violations Make Intentional Misunderstandings Worse.” Cogsci, 2023; with children in Bridgers et al. “Loopholes, a Window into Value Alignment and the Learning of Meaning.” Cogsci, 2021).
>
> Finally, we appreciate your highlighting that the phenomena we study in the evaluation task are nuanced and are potentially sensitive to a variety of cultural and demographic factors, as well as the pointer to those particular supporting works. We look forward to expanding our discussion section to include these citations and to elaborate on why we believe our study is still a fair test of the models’ capabilities based on the cultural context to which they have been most exposed: Namely, the models we test are known to have been trained on text data that reflects a very similar distribution of cultural backgrounds as the population studied in our work. Thus, we do not think that the models’ failure to reason about humor and distress can be attributed to the lack of a nuanced understanding of the cultural variation in these realms, but instead is due to their lack of cognitive understanding of the expectations, social norms, and incongruities needed to reason about these phenomena (Sap et al. “Neural Theory-of-Mind? On the Limits of Social Intelligence in Large LMs.” EMNLP, 2022; Hu et al. “A fine-grained comparison of pragmatic language understanding in humans and language models”. ACL, 2023; Ullman. “Large Language Models Fail on Trivial Alterations to Theory-of-Mind Tasks.” Arxiv, 2023.).
>
> –
>
> Point 2, motivation for models: The models we test represent the current SOTA for the zero-shot, instruction-based prompting that most faithfully replicates the human experiments and that have been used to test pragmatic reasoning in related works (Hu et al., 2023; Yu et al. ALERT: Adapting Language Models to Reasoning Tasks. ACL, 2023; Ruis et al. “Large language models are not zero-shot communicators.” Arxiv, 2022). In the revision, we will emphasize these connections to other related work and the value of testing the same set of models across different reasoning abilities to contribute to a more comprehensive understanding of their strengths and weaknesses in this domain.
>
> While we touch on some effects of model size in relation to GPT-3 and 3.5’s standout performance on the generation task (lines 617-619), we will also expand the discussion to include more observations about the effects of scaling model size and training data in our revision. For example, at 250M parameters, FlanT5-base is significantly smaller than any other model we test, and also performs the worst across both the evaluation and generation tasks. Further, in addition to their smaller parameter size compared to the GPT models, all the T5-based models we test also have smaller data input than the GPT models, with T5’s C4 corpus comprising just 60% of GPT-3’s training data. When comparing the performance of our two families of T5-based models, Tk-Instruct’s fine-tuning on the larger, and more diverse Super-NaturalInstructions benchmark compared to the FlanT5 task set, also appears to translate to improvements on our task. Still, there appear to be interesting exceptions to the benefits of scaling on our task - for example, InstructGPT shows comparable performance to the two Tk-Instruct models on the generation task, despite it being significantly larger than either (175B parameters vs. 3B and 11B). This could be because InstructGPT was specifically fine-tuned to improve alignment and better follow instructions (https://openai.com/research/instruction-following), suggesting that instruction-following fine-tuning may not necessarily improve the flexibility of pragmatic reasoning.
>
> –
>
> Point 3, statistical analysis: We would like to highlight the statistical analyses that we present in Section 3.1, in which we report the results of a paired t-test for each of our significant claims. Other claims in this section for which we did not include statistics have visually non-overlapping error bars (95% confidence intervals) e.g. “GPT-3 and Tk-Instruct-3B also perform reasonably well on this metric, with loopholes resulting in significantly less upset than non-compliance…”, lines 455-457. We appreciate the comments on improving the presentation of our results, and will revise the section to include the confidence intervals and p-values for such claims.
>
> Relatedly, in response to the comment that “Visually, it looks like at least FlanT5-XL and FlanT5-XXL also 'reflect' human ratings if by 'reflect' the authors mean the same sorting order.”, sorting order was not enough; we cared that they significantly differentiated loopholes from non-compliance and understood that compliance resulted in low levels of upset. The previously mentioned lines from section 3.1 discuss the significant difference between Tk-Instruct-3B, GPT3, and GPT3.5’s upset ratings for loopholes vs. non-compliance (i.e., loopholes rated significantly lower than non-compliance), which FlanT5-XL and FlanT5-XXL did not exhibit (i.e. rating of loopholes were not significantly different from non-compliance in a paired t-test; p = 0.19, FlanT5-XL; p = 0.13, FlanT5-XXL). Furthermore, both of the FlanT5 models have significantly higher upset ratings for compliant behaviors than the three models we pull out in our original claim (and these ratings for compliance are also higher than even the non-compliance upset ratings for the humans). We believe these critical differences in how the FlanT5 models vs. the GPT and Tk models compare to human baseline ratings of upset justifies their not being mentioned in said claim. However, from the comment, we recognize that these criteria were not made clear and that the paper would benefit from including these clarifications around how we are characterizing the results and adding the appropriate p-values, etc.
>
> Second, our claim "the advantages of increasing model size to improve pragmatic reasoning abilities are apparent" (lines 617-619) immediately follows a summary of the generation task results which specifically discusses the standout performance of GPT-3 and GPT-3.5 (lines 613-617). Thus, this claim was intended to be considered in reference to that particular task, not the evaluation one, and to those particular models (which are significantly larger than any of the Tk-Instruct or FlanT5 models tested). However, we acknowledge that our claim was vaguely worded and broad enough to warrant this confusion and would benefit from being qualified to the particular family of models whose discussion immediately precedes it. Thank you for highlighting this point.
>
> –
>
> Regarding reproducibility concerns: we would like to mention lines 339-350 which describe settings for the only parameter we manipulated, temperature. We will additionally clarify that all other parameters were set to the model defaults so as to evaluate the out of the box capabilities of these models.

---

### Official Review · Reviewer_4KS3 · 2023-08-04

**Soundness:** 4

**Excitement:**

4: Strong: This paper deepens the understanding of some phenomenon or lowers the barriers to an existing research direction.

**Paper Topic And Main Contributions:**

The paper assesses the capabilities of current LLMs to evaluate and generate loophole behaviour: Humans and LLMs (8, including variants of Tk-Instruct, Flan-T5 and GPT) were presented with scenarios based on real-world anecdotes where a protagonist is asked by another person to pursue or refrain from some activity. Whereby the instructor is in an upwards power relation to the protagonist. In one experiment, the humans (180 individuals resident in the US and fluent in English from diverse regional and educational backgrounds, female and male balanced out, 70% White and the rest a mix of Hispanic, Black, Asian, Mixed and other ethnicity) and the LLMs were presented with 3 types of protagonist behaviour: compliance, non-compliance, loophole, and they were asked to assess the respective behaviours according to a 4-point Likert scale along the dimensions (i) how much trouble the protagonist will get for their action, (ii) how upset the other person will be by the protagonist’s action, (iii) how funny the other person finds the protagonist’s behaviour. The model outputs were then automatically coded in numerical responses on the 4-point Likert scale and manually verified. In the second experiment humans (52, with a similar distribution as in the first experiment) and the LLMs were in addition to the scenario informed that the protagonist does not want to comply with the request and they were asked to generate a loophole behaviour for the protagonist. The human answers and model outputs were then classified by two annotators into loophole, compliance, non-compliance, unclear and other. The results from the human protagonists are compared with the results from the LLMs, revealing in a quantitative manner major differences between LLMs and humans and among LLMs.

**Reasons To Accept:**

Researching the capacities of LLMs to deal with loophole behaviour is a relevant aspect for getting an impression of pragmatics related behaviour of today’s LLMs. Comparing human performance in the evaluation and generation of loophole behaviour with a variety of current LLMs with different (parameter) sizes is a first step to better understanding language behaviours displayed by different LLMs.

**Reasons To Reject:**

I see no weaknesses that would qualify as reasons to reject the paper.

**Reproducibility:**

4: Could mostly reproduce the results, but there may be some variation because of sample variance or minor variations in their interpretation of the protocol or method.

**Reviewer Confidence:**

4: Quite sure. I tried to check the important points carefully. It's unlikely, though conceivable, that I missed something that should affect my ratings.

**Typos Grammar Style And Presentation Improvements:**

* In section 2.3, for the convenience of the reader, add a table listing all LLMs used and their number of parameters.
* Add information on the intercoder agreement for the 2 tasks.

---

> ### Author Rebuttal · Authors · 2023-08-28
>
> Thank you for taking the time to engage with our work, we appreciate the support and are glad to hear you found the work clear and of interest. We appreciate your helpful comments on the grammar and presentation of our paper and look forward to addressing them.

---

### Official Review · Reviewer_4z98 · 2023-08-05

**Soundness:** 3

**Excitement:**

4: Strong: This paper deepens the understanding of some phenomenon or lowers the barriers to an existing research direction.

**Paper Topic And Main Contributions:**

## Topic:

This paper focuses on research on evaluating and generating vulnerability behavior in humans and large language models (LLMs).

## Contributions:

1. The study found that while AI and machine learning vulnerabilities raise concerns, state-of-the-art LLMs have not been specifically evaluated for understanding and generating vulnerable behavior.

2. The research explores by evaluating and generating two tasks. In the evaluation task, models and humans evaluate situations describing different behaviors (compliance, non-compliance, and vulnerability) based on indicators such as trouble, frustration, and humor. The study aimed to understand how well LLMs understood and responded to ambiguous instructions.

3. Generative tasks involve both models and humans generating vulnerability responses given context. The authors compare the performance of different LLMs and humans on these tasks. The results showed that GPT-3.5 came close to human baselines in vulnerability generation, while other models produced responses ranging from non-compliant to incoherent or irrelevant. This study has important implications for understanding the scope and limitations of pragmatic reasoning in LLMs and addressing AI safety issues related to vulnerable behaviors

**Questions For The Authors:**

1. I am sorry, but where are your related works?

2. From 2.4.1: 'The models’ natural language generations were automatically coded into the corresponding numerical response  on the 4-point scale and then verified by human.' Can you expand on how you encode natural language on a 1-4 scale?

**Reasons To Accept:**

1. The whole task of loophole detecting and generating seems interesting and quite novel;

2. The evaluating method is quite insightful: by detecting trouble, frustration, and humor to define the influence of behaviors.

3. Although the experimental method used in the article is relatively simple (hard-prompting), the results seem to be soundness, effective and instructive.

**Reasons To Reject:**

1. The definition of 'loopholes' is not quite clear, and it should be considered via casual inference, rather than a simple phenomenon;

2. The depth of work is not enough. Only superficial phenomena are discussed.

**Reproducibility:**

4: Could mostly reproduce the results, but there may be some variation because of sample variance or minor variations in their interpretation of the protocol or method.

**Reviewer Confidence:**

4: Quite sure. I tried to check the important points carefully. It's unlikely, though conceivable, that I missed something that should affect my ratings.

---

> ### Author Rebuttal · Authors · 2023-08-28
>
> Thank you for taking the time to engage with our work, and for your helpful feedback and comments. We were glad to hear you found the work interesting and novel, and we appreciate your words regarding our methods and results.
>
> Please consider our responses to your other points below:
>
> Point 1, the definition of loopholes: We agree that the definition requires clarification. We rely on previous work in cognitive science, and consider loophole behavior to be 'the intentional misunderstanding of a given request, favoring a less likely though still possible interpretation in the service of one's own goals.' Such a definition is also in line with previous work in law and moral reasoning on people’s understanding of the letter vs. the spirit of the law. We can amend this definition in a revision.
>
> Point 2, depth: Loophole behavior is a complex, but still under-explored phenomenon, across both the cognitive science and computational literatures. We acknowledge that our work is just a first step towards understanding the mechanisms of this important phenomenon, but we believe that it is still a significant one. There has been widespread interest in the language comprehension and generation abilities of LLMs, particularly in terms of their pragmatic reasoning abilities, and our task introduces nuances to this discussion by highlighting an under-explored, but highly-relevant phenomenon in human language use. We believe that designing systematic evaluations of models’ performance on this task in relation to human baselines, as we do in this paper, is a necessary first step towards understanding what we need to target to design better, more flexible language systems. We also believe our work reveals important gaps in the current abilities of these models, and appreciate the opportunity to elaborate on the design implications of our findings.
>
> Additional questions: Regarding related work, we embedded our discussion of related works in the Discussion section (Section 4). We are happy to additionally elaborate on this section in the revision. Regarding the evaluation task (Section 2.4.1), the models’ generations were constrained to the same 4 answer options as those in the human experiment, mentioned in lines 393-396, which correspond to a 4-point Likert: no trouble/not upset/not funny (0), a little bit of trouble/a little bit upset/a little bit funny (1), trouble/upset/funny (2), or a lot of trouble/very upset/very funny (3)). This constraint was enforced by appending the relevant set of valid options to the instruction prompt for each evaluation metric (Section 2.3.2). Generations that didn’t adhere to one of four answer options were considered invalid responses and not included in Figure 2. Nearly all of these invalid responses were empty strings, which is documented in Figure 4 of the Appendix.

---

### Meta-Review · Area_Chair_8dud · 2023-09-12

**Recommendation:** 2

**Metareview:**

Review Summary:

The paper under review investigates the evaluation and generation of loophole behavior in humans and large language models (LLMs). It conducts experiments comparing the performance of different LLMs to humans in assessing and generating behaviors related to compliance, non-compliance, and vulnerability in scenarios with power dynamics. The primary findings indicate disparities between LLMs and humans in understanding and generating these behaviors, with GPT-3.5 performing relatively well in vulnerability generation. The paper addresses a relevant topic, shedding light on the pragmatic reasoning abilities of LLMs and their implications for AI safety. However, there are some concerns regarding methodology, definition clarity, and depth of analysis.

Pros from Reviews:

- Relevant Topic: The paper tackles a timely and important issue concerning the evaluation and generation of loophole behavior in language models, which is crucial for understanding the capabilities and limitations of these models.

- Experimental Insight: The paper provides valuable insights into the differences between LLMs and humans when evaluating and generating behaviors in scenarios with power dynamics, using a well-structured experimental approach.

- Implications for AI Safety: The research has implications for AI safety by highlighting the potential challenges posed by LLMs in understanding and generating vulnerable behaviors.

Cons from Reviews:

- Methodology Concerns: Reviewer 01 raises valid concerns about the methodology, including the need for more details on human evaluation, potential cultural influences on humor and distress, and the lack of statistical analysis for interpreting results. These issues impact the robustness of the study's claims.

- Definition Clarity: Reviewer 03 notes that the definition of 'loopholes' could be clearer and suggests a more in-depth exploration beyond superficial phenomena. A clearer definition and deeper analysis could enhance the paper's quality.

As AC, I have noted the large discrepancy between the soundness scores of the reviewers and asked for a discussion. Only one reviewer gave convincing arguments as to why the soundness is borderline. Looking at the depth of the review by Reviewer HszC and their arguments in comparison with the other reviews, I have to side with Reviewer HszC and agree that at this point there are open questions regarding the methodology which should be revised before the paper can be accepted. Nonetheless, I also agree with the reviewer that "[t]he work tackles an interesting and challenging research question and is worth the attention of the NLP community".

---

### Decision · Program_Chairs · 2023-10-07

**Decision:**

Accept-Findings

**Comment:**

Review Summary:

The paper under review investigates the evaluation and generation of loophole behavior in humans and large language models (LLMs). It conducts experiments comparing the performance of different LLMs to humans in assessing and generating behaviors related to compliance, non-compliance, and vulnerability in scenarios with power dynamics. The primary findings indicate disparities between LLMs and humans in understanding and generating these behaviors, with GPT-3.5 performing relatively well in vulnerability generation. The paper addresses a relevant topic, shedding light on the pragmatic reasoning abilities of LLMs and their implications for AI safety. However, there are some concerns regarding methodology, definition clarity, and depth of analysis.

Pros from Reviews:

- Relevant Topic: The paper tackles a timely and important issue concerning the evaluation and generation of loophole behavior in language models, which is crucial for understanding the capabilities and limitations of these models.

- Experimental Insight: The paper provides valuable insights into the differences between LLMs and humans when evaluating and generating behaviors in scenarios with power dynamics, using a well-structured experimental approach.

- Implications for AI Safety: The research has implications for AI safety by highlighting the potential challenges posed by LLMs in understanding and generating vulnerable behaviors.

Cons from Reviews:

- Methodology Concerns: Reviewer 01 raises valid concerns about the methodology, including the need for more details on human evaluation, potential cultural influences on humor and distress, and the lack of statistical analysis for interpreting results. These issues impact the robustness of the study's claims.

- Definition Clarity: Reviewer 03 notes that the definition of 'loopholes' could be clearer and suggests a more in-depth exploration beyond superficial phenomena. A clearer definition and deeper analysis could enhance the paper's quality.

As AC, I have noted the large discrepancy between the soundness scores of the reviewers and asked for a discussion. Only one reviewer gave convincing arguments as to why the soundness is borderline. Looking at the depth of the review by Reviewer HszC and their arguments in comparison with the other reviews, I have to side with Reviewer HszC and agree that at this point there are open questions regarding the methodology which should be revised before the paper can be accepted. Nonetheless, I also agree with the reviewer that "[t]he work tackles an interesting and challenging research question and is worth the attention of the NLP community".